

# DNA mini-barcoding reveals the mislabeling rate of canned cat food in Taiwan

Yu-Chun Wang[1,2], Shih-Hui Liu[3], Hsuan Ching Ho[4], Hsiao-Yin Su[5] and Chia-Hao Chang[5]

[1] Institute of Fisheries Science, National Taiwan University, Taipei, Taiwan
[2] Technical Service Division, Fisheries Research Institute, Keelung, Taiwan
[3] Department of Biological Sciences, National Sun Yat-sen University, Kaohsiung, Taiwan
[4] Department and Graduate Institute of Aquaculture, National Kaohsiung University of Science and Technology, Kaohsiung, Taiwan
[5] Department of Science Education, National Taipei University of Education, Taipei, Taiwan

Corresponding author
Chia-Hao Chang,
chiahao0928@gmail.com

## ABSTRACT

**Background:** Domestic cats are important companion animals in modern society that live closely with their owners. Mislabeling of pet food can not only harm pets but also cause issues in areas such as religious beliefs and natural resource management. Currently, the cat food market is booming. However, despite the risk that mislabeling poses to cats and humans, few studies have focused on species misrepresentation in cat food products.

**Methods:** To address this issue, we used DNA barcoding, a highly effective identification methodology that can be applied to even highly processed products. We targeted a short segment (~85 basepairs) of the mitochondrial 16S rRNA (*16S*) gene as a barcode and employed Sanger or next generation sequencing (NGS) to inspect 138 canned cat food products in the Taiwanese market.

**Results:** We discovered that the majority of mislabeling incidents were related to replacement of tuna with other species. Moreover, our metabarcoding revealed that numerous undeclared ingredients were present in all examined canned products. One product contained CITES Appendix II-listed shortfin mako shark (*Isurus oxyrinchus*). Overall, we uncovered a mislabeling rate of at least 28.99%. To verify cases of mislabeling, an official standardized list of vernacular names, along with the corresponding scientific species names, as well as a dependable barcoding reference sequence database are necessary.

## INTRODUCTION

Cats (*Felis catus*) are one of the two main pets worldwide. They appear to have been first domesticated ~12,000 years ago, possibly to rid grain stores of rodent pests (*Nilson et al., 2022*). Today, more than half of all households worldwide have a pet (*Alexander et al., 2020*), where they play an important role as emotional company. In modern societies, most owners typically feed their pets with commercial food, so pet food safety is clearly an

important issue. Many animal-sourced proteins can cause food allergies or food intolerances in pets (*Craig, 2019*; *Jackson, 2023*), or they may contain toxins such as scombrotoxin or indospicine (*e.g.*, *Salmon et al., 2022*). What meat is contained in pet food also has religious implications. Pets live so close to their owners that the same tableware may be used for both human and pet food and it may be washed in the same basin. Moreover, pet and human foods may be stored together. Accordingly, if a pet food contains an ingredient specifically prohibited for religious reasons, such as pork for Muslims, contamination is a concern (*Amir et al., 2014*).

Just as extensively described for human foods, pet food also suffers from species misrepresentation, *i.e.*, substitution of one species for another or inclusion of undisclosed species (*Armani et al., 2015*; *Okuma & Hellberg, 2015*; *Cardeñosa, 2019*; *Palumbo et al., 2020*; *Preckel et al., 2021*; *French & Wainwright, 2022*; *Zhu, Alden & Edwards, 2023*). Such false labeling might be prevalent in cat food, especially for cat foods containing fish since fishery products are primary ingredients. Excess demand and over-exploitation (*Ye & Gutierrez, 2017*) both contribute to fishery products being some of the most heavily falsified on the market (*Marvin et al., 2022*). Species misrepresentation has been confirmed for a diverse array of fishery products, including fillets and sushi, as well as canned, roasted and smoked products (*Chang et al., 2021a*; *Panprommin & Manosri, 2022*; *Giusti et al., 2023*; *Kitch et al., 2023*). Whether species misrepresentation is unintentional (owing to species misidentification) or deliberate (driven by financial gain), it can still negatively impact fisheries management, conservation policy, public health, and consumer confidence (reviewed in *Chang et al., 2021b*).

Traditionally, fishery species identification depends on morphological appearance, but food-processing activities (*e.g.*, deheading, scaling, and filleting) often remove these diagnostic characters. Some "micro-characters" can persist after processing, such as the calcareous structures of sea cucumbers (*Aydin & Erkan, 2015*), but this is generally not the case for most fishery products. Nowadays, DNA barcoding has become the "gold standard" for food authentication, which involves identifying a species-specific DNA fragment in a food sample (*Hebert et al., 2003*). Mitochondrial genes are often chosen as genetic markers for species identification in food samples because of their large copy number in cells, with 12S rRNA (*12S*), 16S rRNA (*16S*), cytochrome b (*Cyt b*), and cytochrome c oxidase subunit I (*COI*) all being widely employed (*Fernandes, Amaral & Mafra, 2021*). The molecular identification of food samples often encounters two main problems. First, some food processing activities, such as canning, degrades DNA into short fragments, with one solution being to employ DNA mini-barcoding on fragments of <300 basepairs (bp) (*Frigerioa et al., 2021*; *Preckel et al., 2021*; *Mottola et al., 2022b*; *Roungchun, Tabb & Hellberg, 2022*). Second, food products often comprise more than one species, so conventional Sanger sequencing on such samples typically suffers from signal disruption. Accordingly, metabarcoding can be deployed, which utilizes universal primers so that essentially all candidate species in a test sample can be uncovered (*Galimberti et al., 2019*; *Fanelli et al., 2021*).

*De Silva & Turchini (2008)* estimated that canned cat food accounts for ~6% of global wild fish catch and this industry is booming, with Mordor Intelligence estimating a

compound annual growth rate of 5.3% for 2022–2027 (https://www.mordorintelligence.com/industry-reports/global-cat-food-market-industry). Given the significant consumption of fish by the cat food industry and growing demand, cat food mislabeling investigations are warranted, otherwise cat health could be jeopardized and a loophole in management policies for the sustainable use of fishery resources could arise. Nevertheless, to date, molecular authentication studies on cat food products worldwide remain limited (*Armani et al., 2015*; *Okuma & Hellberg, 2015*; *Günther, Raupach & Knebelsberger, 2017*; *Cardeñosa, 2019*; *Palumbo et al., 2020*; *Dunham-Cheatham et al., 2021*; *French & Wainwright, 2022*). In their study of canned tuna products conducted in Taiwan, *Chang et al. (2021a)* included canned cat food samples, but they only examined 25 products. Thus, the objective of this study was to conduct a thorough survey of canned cat food products available in the Taiwanese market to gain a better understanding of the prevalence of species misrepresentation.

## MATERIALS AND METHODS

### Sample collection

We purchased a total of 138 canned cat food products, belonging to 62 brands, from pet stores in Taiwan. Information on brand, manufacturer or importer, place of manufacture, labeling, and ingredients, which were typically written in Chinese, were all recorded. If the cans had English labels, this data was also recorded (Table S1). All cans were photographed using a Sony α6400 camera with an 18-105 F4 lens (Information S1). If a canned cat food product contained solid meat, such as chunks or flakes, a small quantity of each type of meat (defined according to texture and color) was removed using autoclaved dissection tools, washed with 95% ethanol, and then preserved in 99.5% ethanol at −20 °C until DNA extraction. For minced or paste-like products for which it was impossible to isolate a piece of meat from the homogeneous contents, a small sample was taken at random from the tin and preserved at −20 °C until DNA extraction. Overall, we obtained a total of 261 samples from our collection of canned products.

### Common names and corresponding species reference list

All the sampled products had their ingredients listed with common names instead of scientific species names. Unfortunately, there is no official standardized list of vernacular names and corresponding scientific names available yet. To address this, we compiled a reference list by consulting various resources, including A Guide Book of Common Economic Aquatic Animals and Plants in Taiwan (*Shao et al., 2015*), the fish database of Taiwan (https://fishdb.sinica.edu.tw/), Breed Resources for Livestock Industry (*Lai et al., 2004*), and our previously published barcoding article on canned tuna (*Chang et al., 2021a*). Some of the imported products displayed labeling both in Chinese and the language of source, but we exclusively relied on the Chinese label in these cases since Chinese is the only official language of Taiwan. Our reference list is presented in Table 1.

**Table 1 Summary of Chinese and English labels, English translations, and corresponding species for all collected canned cat food products.**

| Ingredient no. | Declared ingredient in Chinese | English equivalent | Corresponding species |
|---|---|---|---|
| Fish | | | |
| 1 | 鯷魚 | Anchovy | Engraulidae |
| 2 | 巴沙魚 | Basa | *Pangasius bocourti* |
| 3 | 蘭鱈魚 | Blue whiting | *Micromesistius* spp. |
| 4 | 鰹魚, 白鰹魚 | Bonito | *Katsuwonus pelamis, Auxis* spp., *Euthynnus* spp., and *Allothunnus fallai* |
| 5 | 魚子醬 | Caviar | Acipenseridae |
| 6 | 丁香魚 | Clove fish | *Spratelloides gracilis* |
| 7 | 鱈魚 | Cod | Gadiformes |
| 8 | 魴魚 | Dory | Zeidae |
| 9 | 魚油 | Fish oil | Fish |
| 10 | 比目魚 | Flatfish | Pleuronectiformes |
| 11 | 柴魚, 柴魚片, 正鰹 | Katsuobushi | *Katsuwonus pelamis* |
| 12 | 白身鮪魚, 鮪魚白肉 | Light tuna | *Thunnus albacares* and *Katsuwonus pelamis* |
| 13 | 鯖魚, 青花魚 | Mackerel | *Scomber* spp. |
| 14 | 旗魚 | Billfish | Istiophoriformes |
| 15 | 虱目魚 | Milkfish | *Chanos chanos* |
| 16 | 秋刀魚 | Pacific saury | *Cololabis saira* |
| 17 | 鮭魚 | Salmon | Salmonidae |
| 18 | 沙丁魚 | Sardine | *Amblygaster* spp., *Dussumieria* spp., *Etrumeus* spp., *Sardinella* spp. |
| 19 | 鯛魚 | Sea bream | Berycidae, Priacanthidae, Lobotidae, Lutjanidae, Haemulidae, Glaucosomatidae, Sparidae…[*] |
| 20 | 吻仔魚 | Whitebait | Clupeiformes |
| 21 | 鮪魚, 鮪魚(鰹魚) | Tuna | *Thunnus* spp. and *Katsuwonus pelamis* |
| 22 | 白身魚 | White fish | Any white-fleshed fish with a mild flavor |
| 23 | 黃鰭鮪魚 | Yellowfin tuna | *Thunnus albacares* |
| 24 | 蟹柳, 蟹肉棒, 蟹味絲 | Kanikama | Fish |
| Poultry | | | |
| 25 | 雞肉, 雞胸肉 | Chicken | *Gallus gallus* |
| 26 | 火雞肉 | Turkey | *Meleagris* spp. |
| Livestock | | | |
| 27 | 牛肉, 牛肉絲 | Cattle | *Bos* sp. and *Bubalus bubalis* |
| 28 | 鹿肉 | Deer | Cervidae |
| 29 | 羊肉, 羔羊血, 羊肚 | Sheep/goat | *Capra hircus* and *Ovis aries* |
| Reptiles | | | |
| 30 | 鱉肉, 鱉蛋粉 | Softshell turtle | Trionychidae |
| Crustaceans | | | |
| 31 | 櫻花蝦 | Sergestid shrimp | *Sergia lucens* |
| 32 | 蝦, 鮮蝦, 蝦肉, 鮮蝦仁 | Shrimp | Shrimp |
| 33 | 蟹肉 | Crab | Crab |
| Mollusks | | | |
| 34 | 蛤蜊 | Clam | Veneridae |

| Ingredient no. | Declared ingredient in Chinese | English equivalent | Corresponding species |
|---|---|---|---|
| 35 | 黃金蜆 | Freshwater clam | *Corbicula fluminea* |
| 36 | 綠貽貝, 綠唇貽貝 | Green mussel | *Perna viridis* |
| 37 | 干貝 | Scallop | Pectinidae |
| 38 | 魷魚 | Squid | Squid |

**Note:**
[*] 鯛 is an umbrella term. Except for the cichlid fishes, the Chinese names of perciform fishes (according to the Taiwan Fish Database) ending with "鯛" may generally be denoted by 鯛魚.

## Molecular identification

DNA extraction Kit S (Cat No./ID: GS100; Geneaid, Taipei, Taiwan) was employed to extract DNA from each of the 261 samples. To amplify the mitochondrial *16S* rRNA fragment (*16S*) (~85 bp), PCR was performed in a mixture containing 20 ng template DNA, 50 µL of 2× Taq PCR MasterMix (GN-PCR201-01; Genomix, Noida, Uttar Pradesh), and 50 µmol of each forward and reverse primer. We designed forward and reverse primers specific to *16S* (*Horreo et al., 2013*), with the addition of M13 primers to facilitate Sanger-based sequencing—Forward, M13F(-20)16S-HF (5′-GTAAAACGAC GGCCAGTATAACACGAGAAGACCCT-3′); Reverse, M13R(-24)16S-HR1+2 (5′-AACAGCTATGACCATGCCCRCGGTCGCCCCAAC-3′)—or by adding one Illumina index sequence (D701 to D712) to the forward primer for further metabarcoding next-generation sequencing (NGS) analysis. The M13-modified primers were used for DNA samples extracted from solid meat, whereas the index-modified forward primer pair was employed for other sample types. These primers were made up to a final volume of 100 µL using distilled water. Thermal cycling began with one cycle at 95 °C for 4 min, followed by 35 cycles of denaturation at 95 °C for 30 s, 50 °C for 30 s, and 72 °C for 30 s and, finally, a single extension step at 72 °C for 7 min. A negative control of sterile water instead of a DNA sample was also included with each PCR reaction. All necessary procedures were adopted to eliminate DNA contamination in the laboratory. PCR products were purified using a PCR DNA Fragment Extraction Kit (Geneaid, Taipei, Taiwan).

The fragments amplified by M13-modified primers were subjected to Sanger sequencing, performed by Mission Biotech (Taipei, Taiwan), using M13 sequencing primers. Primer sequences linked to the amplified fragments were trimmed before constructing contigs using CodonCode Aligner. The fragments amplified by the index-added forward primer pair were subjected to NGS using an Illumina system. The purified PCR products were quantified using an Invitrogen Qubit 4 Fluorometer. Ten to 11 PCR products from each sample were pooled and then prepared to produce a single DNA library. Illumina sequencing to generate 150-bp paired-end reads was conducted on a NovaSeq 6000 platform by Genomics Biotech (New Taipei City, Taiwan). Approximately 1 GB output was generated for each sample. Illumina reads were trimmed using Trimmomatic 0.39 (*Bolger, Lohse & Usadel, 2014*) to remove Illumina indexes, the

index-added forward primer sequences, and low-quality bases. The trimmed reads were then paired using FLASH 1.2.11 (*Magoč & Salzberg, 2011*). Paired reads were clustered into an operational taxonomic unit (OTU) using USEARCH 11.0.667 (*Edgar, 2010*) if they were 100% identical. The coverage of each OTU was also determined. The NGS raw reads have been submitted to GenBank with BioProject number PRJNA1036020. Sanger-sequenced *16S* rRNA fragments of <200 bp cannot be submitted to GenBank (https://www.ncbi.nlm.nih.gov/genbank/submit_types/), but we include them in Information S2. All query *16S* rRNA segments generated in this study are accessible at Figshare (https://figshare.com/articles/dataset/16S_mini_barcoding_results_of_138_cat_cans/22097792).

## Data analysis

Species identifications of newly generated *16S* sequences were achieved by comparing them (by BLAST) to reference sequences in the NCBI GenBank database using Geneious Prime® 2023.0.4 software (Biomatters Ltd., Auckland, New Zealand). For Illumina data, species identifications were only conducted on OTUs with a coverage of at least 10. Following previously published approaches (*Armani et al., 2015*; *Horreo et al., 2019*; *Mottola et al., 2022a*), we first defined that only matches displaying full sequence coverage, 100% similarity, and with unambiguous species-level scientific names were considered positive fish identifications. If no BLAST matches displayed 100% similarity, then matches with >98% similarity (allowing 1 bp mismatch or gap) were deemed acceptable. When more than one fish species was revealed as a positive match, all of them were considered potential candidates (Table 1).

## Comparison of molecular identification results and product labels

We checked if the declared ingredients of a canned food sample were also identified by molecular identification. We ran a binominal model in the "glm" function of the "lm4" package in R version 3.6.1 (*R Core Team, 2019*) to test if the declared ingredients or place of manufacture influenced the success of molecular detection. This model can be summarized as follows: positive or negative molecular detection ~ a declared ingredient item + the place of manufacture. Then, a cat food product was judged as mislabeled if the molecular authentication result did not match the ingredient list, whether that meant that the molecularly-identified species was not present in the ingredient list or the fish species expected in the sample based on the ingredient list was absent. If DNA authentication failed to identify a species from a sample, that sample was excluded from our assessment of product mislabeling. If all samples from a canned product failed molecular authentication, the product was assigned as uncertain. Another binominal model was used to establish if the place of manufacture influenced the mislabeling of a cat food product. Since the initial model suffered from over-dispersion, the observation-level was included as a random effect (*Harrison, 2014*). This model was run using the "glmer" function of the "lm4" package in R version 3.6.1 and can be summarized as: whether or not the product is mislabeled ~ the place of manufacture + (1|No. of products).

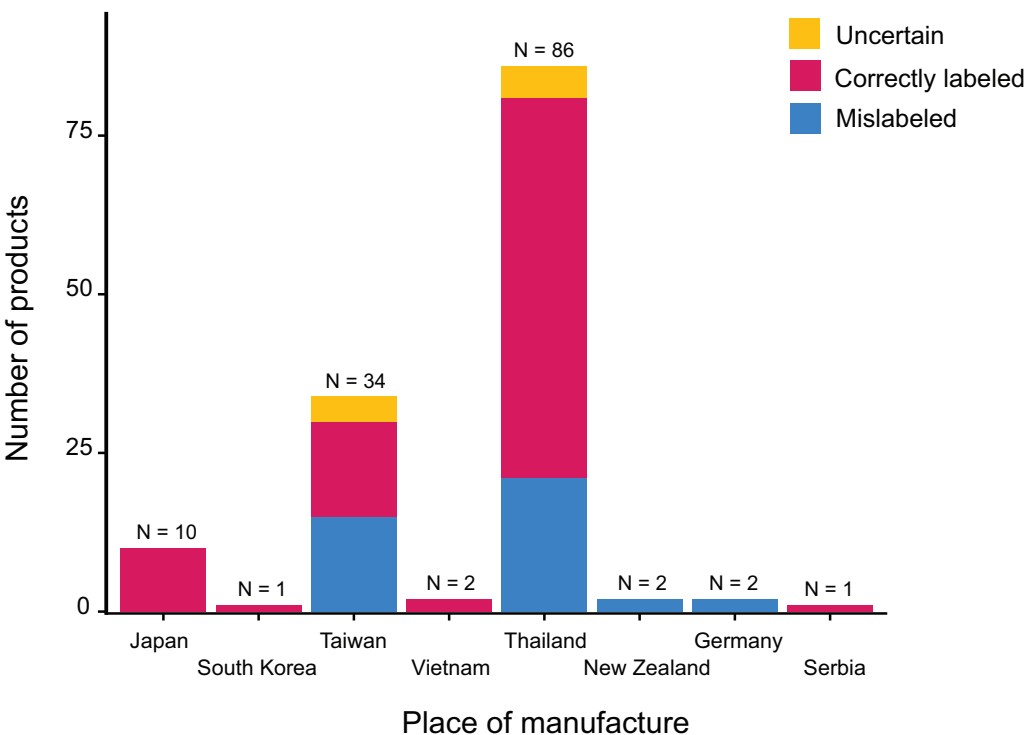

**Figure 1 Place of manufacture for the 138 collected canned cat food products and associated *16S* mini-barcoding results.**

## RESULTS

The 138 canned cat food products we tested in this study were manufactured in eight different countries, though most of them were produced in either Thailand or Taiwan (Fig. 1). Collectively, we identified 38 different labeled ingredients (Table 1 and Fig. 2): 24 fish, two poultry (chicken and turkey), three livestock (sheep/goat, cattle, and deer), one reptile (softshell turtle), three crustaceans, and five mollusks. Tuna and chicken proved to be the most common cat food ingredients ($n = 67$ and 48, respectively) (Fig. 2).

Of the 261 DNA samples successfully extracted from the canned products, 240 underwent Sanger sequencing after successful PCR, but *16S* amplicons could only be generated for 182 of these latter. A further 21 DNA samples were successfully sequenced using the Illumina platform. NovaSeq generated 2,165,707 paired reads for 4,667 OTUs. The overall sequencing success rate of the study was 77.78% (182 + 21/261) (Table S1).

Our BLAST analysis successfully identified a source species for all 203 samples for which we had generated *16S* amplicons. For the 138 investigated canned products, nine could be categorized as uncertain, 89 were deemed correctly labeled, but the remaining 40 products were mislabeled, *i.e.*, either the molecularly-identified species was not the species expected based on the declared ingredients or the molecularly-identified species was absent from the ingredient list. The *16S* mini-barcoding analysis using our Sanger sequencing results on 19 of the mislabeled products revealed that in many cases ($n = 9$) products labeled as containing tuna actually do not have any tuna (*Thunnus* spp. or *Katsuwonus pelamis* (skipjack)), instead comprising *Auxis* spp. or *Euthynnus affinis* (Table S1). NGS
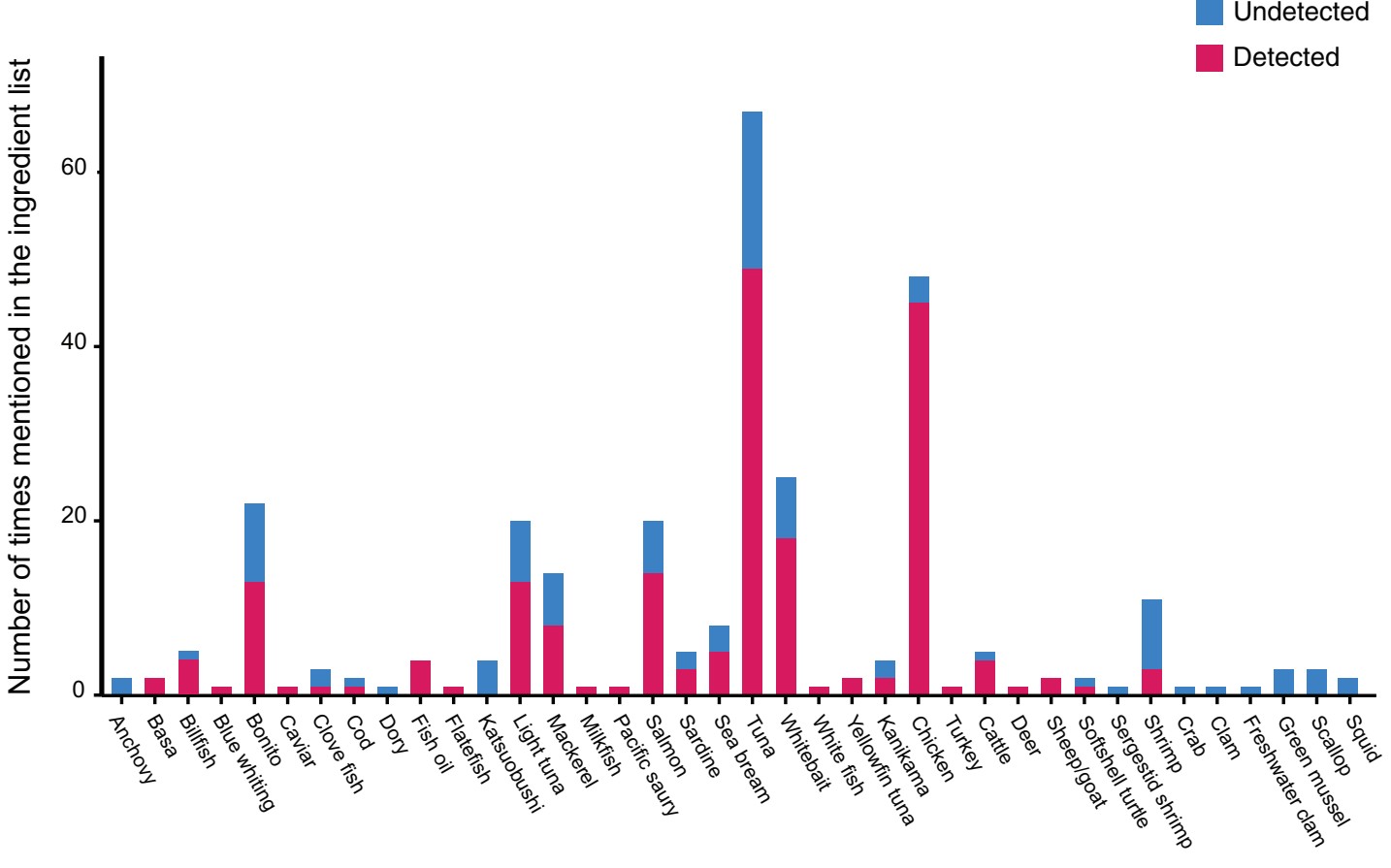

**Figure 2** Number of times 38 ingredients (see details in Table 1) were listed on labels of the collected canned cat food products, and whether or not they were detected by *16S* mini-barcoding.

metabarcoding of the other 21 mislabeled products revealed a highly diverse list of ingredients, many of which were undeclared on the labels, including cattle, deer, pig, sheep/goat, turkey, quail, shrimp, and many fishes. Overall, the mislabeling rate of canned cat food products in the Taiwanese market is at least 28.99% (40/138).

Our binominal model revealed that the probability of a given ingredient being detected by our *16S*-based metabarcoding authentication approach varies significantly across food types ($\chi^2 = 95.444$, $p < 0.01$) (Fig. 2), with tuna and chicken frequently being verified molecularly, but none of the mollusks (clam, mussel, scallop, and squid) apparently occurring in any of the canned products despite being labeled as ingredients. Statistical analysis also demonstrated that mislabeling rates are not significantly biased by where the canned products are manufactured ($\chi^2 = 4.9673$, $p = 0.66$) (Fig. 1).

## DISCUSSION

Apart from two types of poultry meat (chicken and turkey) and three of livestock (beef, venison, and lamb), fishery products, and primarily fish, proved to be the major

ingredients of canned cat food in Taiwan (Fig. 2). Among the fishes used as cat food, Thunnini encompassing the genera *Thunnus*, *Katsuwonus*, *Auxis*, *Euthynnus*, and *Allothunnus* accounted for the majority of cases. This outcome is not surprising since these latter are responsible for ~10% of the global seafood trade, with ~75% of Thunnini catch destined for the canning industry. Thailand is the largest exporter globally of canned Thunnini (*Brill & Hobday, 2017*; *Guillotreau et al., 2017*; *Mata et al., 2020*), explaining why most of the canned cat food products we examined are manufactured in Thailand, even though many are Taiwanese brands, and Thunnini-related fishes were common among the respective ingredients lists (Figs. 1 and 2). Some fish species belonging to Thunnini are commonly referred to as tuna, but only *Thunnus* and *Katsuwonus* fishes can legally be used worldwide to produce commercial tuna products. Mislabeling of tuna products (*i.e.*, replacement with other Thunnini species) is a serious problem in the seafood industry as it deceives consumers (reviewed in *Chang et al., 2021a*). Our study conducted in Taiwan uncovered that at least 28.99% of the canned cat food products we sampled were mislabeled, with "true" tuna frequently being replaced with other Thunnini species, particularly of the genera *Auxis* and *Euthynnus*. The country of origin mentioned on labels is crucial to consumer perceptions (*Charlebois et al., 2016*), but we show that the mislabeling rate is not related to where the products were manufactured, although we acknowledge that the sample size for some countries is limited.

Apart from species substitutions, sampling error (especially given that only one tissue sample of each type of meat in a can was investigated), low DNA quality, and/or ineffective primers could also result in many declared ingredients not occurring in our samples (Fig. 2). Both food processing and the presence of other ingredients can erode DNA quality (*Quinteiro et al., 1998*; *Sajali et al., 2018*). For example, among our samples, only 50% of cat food cans labeled as katsuobushi (in Chinese: 柴魚; samples C1F2, C2D2, E1F2, and E3B3), representing skipjack tuna fillets subjected to simmering, smoking, and fermentation, could be molecularly identified successfully. The efficacy of the modified pair of *16S* primers we utilized has been verified in many other DNA barcoding projects on processed foods (*Günther, Raupach & Knebelsberger, 2017*; *Chang et al., 2021a*; *Chaora et al., 2022*). Our study found that using a short *16S* segment was effective in amplifying 77.78% of our DNA samples, enabling identification of these highly processed food samples. However, using a short *16S* segment comes at the cost of obtaining less information for species-level identification. The limitations of using such a short mitochondrial gene segment as a barcoding marker make it difficult to achieve clear species-level identification when dealing with genetically closely-related or hybrid species. This scenario likely explains why our *16S* mini-barcoding analysis of canned tuna-labeled (*e.g.*, C2B1, C4C1, and J1C1), mackerel-labeled (*e.g.*, C3C1, C3E1, C3E2, and C4D), sardine-labeled (*e.g.*, C3B1, D2C2, and F5A2), billfish-labeled (C6A1, E2A, and I3B2), and caviar-labeled (E3E2) products did not identify particular species (*Viñas & Tudela, 2009*; *Little, Lougheed & Moyes, 2010*; *Bronzi, Rosenthal & Gessner, 2011*; *Chan et al., 2019*; *Mottola et al., 2022b*).

In addition to the limitation of short barcoding segments, ambiguous species identification also arises from the presence of questionable reference sequences in

GenBank. Whenever misidentification or typographical error results in incorrect species names being associated with reference sequences, then any database will present incorrect authentications (*Kappel & Schröder, 2020*; *Fernandes, Amaral & Mafra, 2021*). Our study suffers from this issue of ambiguous reference sequences in Genbank. For instance, the source species of samples B1A and C2C2 are probably *Decapterus russelli* and *Pholis fangi*, respectively, because their BLAST-matched sequences are LC646869 (*Kimura, Takeuchi & Yadome, 2022*) and KC748080 (*Kwun & Kim, 2013*), both derived from reliable systematic studies. *Chang et al. (2021a)* mentioned that accession KM055376 in GeneBank is more likely to be skipjack tuna rather than yellowfin tuna since the sequence is extremely similar to several other reliable skipjack tuna sequences. Likewise, we assert that accession KM198903 is *A. thazard* or *A. rochei* since it is 100% identical to the sequences of *A. thazard* or *A. rochei* generated by *Catanese, Infante & Manchado (2008)*, and MW595783 and LN558771 are *Euthynnus affinis* since they are >99.5% identical to the sequence of *E. affinis* generated by *Iwasaki et al. (2013)*. Moreover, KM198901 is *Lates calcarifer* since this sequence is 100% identical to the sequence of *L. calcarifer* generated by *Domingos, Zenger & Jerry (2015)*. Furthermore, we found that BLAST hits for our sample C2C2 are mainly *Pangasianodon hypophthalmus* (37/49) rather than *Pangasius* spp., so C2C2 is more likely derived from *P. hypophthalmus*. Without a dependable barcoding reference database, it becomes challenging to prosecute mislabeling crime. Accordingly, a reliable database for authentication is sorely needed (*Shehata et al., 2018*; *Mitchell et al., 2019*; *Chang et al., 2021b*).

*Kitch et al. (2023)* mentioned that use of inappropriate market names is one reason for seafood products being mislabeled. Unlike in the European Union and the USA, where there are official standardized lists of vernacular names for species used in foods and their corresponding scientific names (*Vandamme et al., 2016*; *Armani et al., 2017*; *Günther, Raupach & Knebelsberger, 2017*; *Pardo & Jiménez, 2020*; *Kitch et al., 2023*), Taiwanese authorities do not publish standard lists of vernacular names used in the food industry. Instead, many "umbrella" terms are used in the Taiwanese market. For example, for Taiwanese customers, the iconic species represented by the character 鯛 is red sea bream (*Pagrus major*), yet 鯛 is an umbrella term for many types of marine fishes such as the Family Lutjanidae (in Chinese: 笛鯛科), Family Priacanthidae (in Chinese: 大眼鯛科), and Family Sparidae (in Chinese: 鯛科). Generally, fishes with 鯛 in the Chinese name can be denoted by "鯛魚". Moreover, we suggest that the Family Cichlidae (in Chinese: 慈鯛科) should not be labeled as 鯛 even though tilapia is frequently substituted for snapper and sea bream (*Hu et al., 2018*; *Spencer et al., 2020*). The term "beef" (in Chinese: 牛肉) is used as an umbrella term, since it lacks a clear official definition. 牛肉 could refer to meat from species in the Bovidae (in Chinese: 牛科), including Bovinae (in Chinese: 牛亞科), Bovini (in Chinese: 牛族), or *Bos* (in Chinese: 牛屬). Overall, such usage of umbrella terms in labeling leaves huge potential for food falsification (*Giagkazoglou et al., 2022*; *Sharrad et al., 2023*), and thus an official standardized list of vernacular and scientific species names is needed for Chinese-speaking regions and countries (*Xiong et al., 2016*; *Xiong et al., 2019*; *Chang et al., 2021b*; *Neo, Kibat & Wainwright, 2022*).

We found that 19 out of the 117 products we investigated by Sanger sequencing were mislabeled. Moreover, all of the 21 additional canned products we examined by NGS metabarcoding can be deemed as mislabeled because they include species not declared on the labels. Such undeclared species can have different origins, potentially being meat-derived additives (*e.g.*, animal fat and gelatin) (*Okuma & Hellberg, 2015*; *Amqizal et al., 2017*), or merely reflect contamination along the production line. Importantly, our discovery of human, cat, dog, and bacterial *16S* barcodes in our samples (*e.g.*, B2C, F2C, and L3C) indicates that the integrity of cat food production lines may be corrupted. Moreover, inclusion of undeclared species in foods destined for cats is a clear health risk. For instance, beef and chicken are common allergens inducing cutaneous adverse food reactions (CAFRs) in cats (*Mueller, Olivry & Prélaud, 2016*), and they were undeclared ingredients in some of our tested products (*e.g.*, B2C, F3C, and L1B). Similarly, undeclared *Thunnus* in certain other products (*e.g.*, F2C, F3A, and F3C) raises the concern of mercury poisoning from canned cat foods (*Kumar, 2018*; *Dunham-Cheatham et al., 2019*). Having undeclared species in cat food can also be problematic for owners in terms of their religion. Pig was not listed among the ingredients of any of our canned products, but our barcoding analysis identified it in some products (*e.g.*, F2E, F3A, L2D, and L3C), so such products are anathema for Muslim owners.

We detected a non-kosher fish, *i.e.*, shark, in a pet food sample (F2A), which is a cause for concern for Jewish pet owners. This is not the first time that sharks have been detected in pet food samples (*Cardeñosa, 2019*; *French & Wainwright, 2022*). However, compared to these two previous studies that frequently found sharks in the pet food samples they investigated, our study only encountered the endangered shortfin mako shark (*Isurus oxyrinchus*) in one sample. It is not clear why so few shark specimens were found among the samples in the current study, potentially being attributable to a variety of factors such as sample size, the locations where the pet food samples were manufactured, and when the samples were purchased. Despite so little shark flesh being found in the current study, the discovery herein of shortfin mako shark, a CITES Appendix II-listed shark, as well as similar findings from other studies, highlight that the pet food industry should be examined critically as a consumer of global shark catch, especially given rapidly declining shark populations.

Metabarcoding is a powerful tool that can be used to identify the species present in canned cat food samples. However, it cannot accurately determine the exact proportions of each species. The discovery of several undeclared ingredients in canned cat food products raises concerns about the accuracy of declared ingredient proportions. This issue not only warrants further investigation, but also highlights the need to revise current regulations regarding the declaration and proportion of ingredients in canned cat food products. Finally, it is noteworthy that our metabarcoding study demonstrates that undeclared species in canned cat food products is so prevalent. Owing to limited tissue sampling of the other 117 canned products inspected by traditional Sanger sequencing, it is highly possible that all of our investigated products are mislabeled.

# CONCLUSIONS

Our study and those of others (*Okuma & Hellberg, 2015*; *Palumbo et al., 2020*; *Dunham-Cheatham et al., 2021*; *Preckel et al., 2021*; *Zhu, Alden & Edwards, 2023*) support that DNA metabarcoding-based identification is a powerful tool for authenticating pet foods, but the lack of an official standardized list of vernacular names and the corresponding scientific species names, as well as deficiencies in barcoding reference sequence databases, likely impede efforts to verify cases of mislabeling. In our study, we assessed 138 canned cat food products and uncovered a mislabeling rate of 28.99% (40/138), but all of the products investigated by NGS metabarcoding contained undeclared species. Thus, it is highly probable that canned cat food products in the Taiwanese market are mislabeled even more extensively than appreciated. Undisclosed species in canned cat food raises concerns about pet health, as some ingredients can cause food allergies or intolerances. For customers, the presence of undisclosed species in cat foods is not only fraudulent, but also increases the risk of violating religious taboos. We also uncovered that nearly all fishes identified in our samples are likely wild-caught rather than being raised in fish farms. Establishing a more precise picture of resource exploitation is a goal of fishery product barcoding studies. The extensive utilization of Thunnini in cat foods, as well as our detection of an endangered shark in a canned product, demonstrates that further DNA barcoding projects on cat food products are needed. Such studies would not only help protect owners and their cats, but also provide important information for better managing marine fishery resources.

# ACKNOWLEDGEMENTS

The corresponding author would like to acknowledge Ms. Ting-Ting Huang for technical assistance and Dr. John O'Brien for editing.

## Funding

Chia-Hao Chang was supported by grants from the National Science and Technology Council, Taiwan (110-2621-B-152-001 and 111-2621-B52-001-MY2). The funders had no role in study design, data collection and analysis, decision to publish, or preparation of the manuscript.

## Grant Disclosures

The following grant information was disclosed by the authors:
National Science and Technology Council, Taiwan: 110-2621-B-152-001, 111-2621-B52-001-MY2.

## Competing Interests

The authors declare that they have no competing interests.

## Author Contributions

- Yu-Chun Wang conceived and designed the experiments, performed the experiments, prepared figures and/or tables, authored or reviewed drafts of the article, and approved the final draft.
- Shih-Hui Liu analyzed the data, prepared figures and/or tables, and approved the final draft.
- Hsuan Ching Ho analyzed the data, prepared figures and/or tables, and approved the final draft.
- Hsiao-Yin Su performed the experiments, analyzed the data, prepared figures and/or tables, and approved the final draft.
- Chia-Hao Chang conceived and designed the experiments, performed the experiments, analyzed the data, prepared figures and/or tables, authored or reviewed drafts of the article, and approved the final draft.

## Data Availability

The data is available at Figshare: Chang, Chia-Hao (2023). 16S mini barcoding results of 138 cat cans. figshare. Dataset. https://doi.org/10.6084/m9.figshare.22097792.v1.

## Supplemental Information

Supplemental information for this article can be found online at http://dx.doi.org/10.7717/peerj.16833#supplemental-information.

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
