# Peer review of "DNA mini-barcoding reveals the mislabeling rate of canned cat food in Taiwan"

_PeerJ, doi:10.7717/peerj.16833_

## Round 0.1 · original submission · Major Revisions

Dear authors,

Following a thorough review by two esteemed reviewers, it is gratifying to note that both concur on the manuscript's intriguing nature and its merit for publication. Nevertheless, significant revisions are deemed essential to ensure its acceptance.

Best regards,

Armando Sunny

Reviewer 1 ·

Basic reporting

The article is well written, is an easy read and does not over interpret/oversell the results. I find the topic interesting and it will likely appeal to the wide readership of PeerJ. PeerJ is a suitable journal for the publication of work such as this.

I only have very minor comments that I would like the authors to consider addressing

Line 46 – I assume this should be “United”, not “Unite”?

Line 116 – “GüntherRaupach & Knebelsberger” I think this should be “Günther, Raupach & Knebelsberger T” note the comma in the correction after Günther. There are several other examples of this throughout the MS e.g., between lines
305 – 308 and others.

Line 344 – Consider changing “Pig was never listed” to “Pig was not listed”

Line 347 – Consider toning down the language regarding the finding of a shark, while I agree it is not good given the declines in shark population numbers. I think finding one shark in all the samples is not really disturbing, same with the instance in the abstract (Worryingly, we detected…).

Related to the above, it might be worth commenting on possible reasons pet food in the US and Singapore contained so much more shark meat and had a greater diversity of species - I don’t know the answer to this, so mild speculation is fine. Maybe supply chains, regulations etc. I’m actually surprised you only found one shark in all those samples, especially given the results of previous work in the US and Singapore (this is a good thing though)

Do check this, but I think the species of shark you found is CITES listed/regulated, this is worth briefly mentioning.

Table 1 is huge, I think the SI section is the most appropriate place for this.

Experimental design

The methods are clearly defined, appropriate and I would be confident of replicating the work based upon what is written.

Validity of the findings

The findings are valid and relevant in context of the data collected.

Reviewer 2 ·

Basic reporting

Overall, the submitted manuscript meets the criteria for basic reporting. Specific comments are provided in the Additional Comments section.

Experimental design

Overall, the submitted manuscript meets the criteria for experimental design. Specific comments are provided in the Additional Comments section.

Validity of the findings

Overall, the submitted manuscript meets the criteria for validity of the findings. Specific comments are provided in the Additional Comments section.

Additional comments

The submitted manuscript, "DNA mini-barcoding reveals the mislabeling rate of canned cat food in Taiwan", focused on investigating the presence and absence of declared fish-based ingredients in canned cat foods sold in Taiwan, and report a near 30% mislabeling rate for these products. Overall, the experimental design and methods are valid, and the interpretation and conclusions are supported by the presented data. However, there are several areas that would make this manuscript more robust, and specific comments are listed below.

The introduction is largely irrelevant to the scope of the work and does not set up the importance of filling the knowledge gap of mislabeling in cat-based wet foods in Taiwan. Many of the paragraphs are unrelated to this topic, and should be omitted and replaced with more relevant supporting information. I suggest the authors develop the main points of the study, and focus the introduction on supporting information leading into the study objective.

The current objective (lines 119-120), "...promoting greater public awareness of [the issue of mislabeling] and its implications", cannot be supported by the experimental design nor results. The study objective should be reconsidered and written in a way that the experimental methods can generate data that can either support or refute the objective/hypothesis.

The discussion section is difficult to follow. Again, I encourage the authors to develop the main points of the study, and focus on those and building the discussion around those points. The way the manuscript is currently written feels haphazard and lacking structure and direction.

Lastly, statements like "hampers prosecution of mislabeling cases" lends to the feeling that this work was performed with an agenda, and that the data may be biased as a result. Be mindful to present the facts and science, and only the facts and science.

Specific comments:
Line 82: "makers" should be "markers"
Lines 321-322: This claim needs more support.

---

## Round 0.2 · accepted · Accept

Dear Authors,

I am pleased to inform you that both reviewers have concurred on the appropriateness of the corrections implemented in your manuscript. As a result, it is with great satisfaction that I announce the acceptance of your work for publication in PeerJ. I extend my heartfelt gratitude for the meticulous attention to detail demonstrated in your revisions.

Thank you for selecting PeerJ as the platform for sharing your compelling and valuable research.

Warm regards,

Armando Sunny

Reviewer 1 ·

Basic reporting

The article has been improved from the original submission and I thank the authors for making the requested changes/edits.

Experimental design

The methods remain well defined, appropriate and I would be confident of replicating the work based upon what is written.

Validity of the findings

The findings remain valid and relevant in context of the data collected.

Reviewer 2 ·

Basic reporting

The revised manuscript is much improved over the initially submitted draft, and meets the journal criteria for basic reporting.

Experimental design

The experimental design is valid and supports the study objective.

Validity of the findings

The findings are supported by the experimental design and not overly interpreted.

Additional comments

Overall, the revised manuscript is much improved from the initially submitted draft. The Introduction and Discussion sections now much more closely support the experiment and findings, and are easy to follow.

Minor comment:
Line 22: "nature" should be "natural"